# Twist and Turn—Topoisomerase Functions in Mitochondrial DNA Maintenance

**DOI:** 10.3390/ijms20082041

**Published:** 2019-04-25

**Authors:** Steffi Goffart, Anu Hangas, Jaakko L. O. Pohjoismäki

**Affiliations:** Department of Environmental and Biological Sciences, University of Eastern Finland, 80101 Joensuu, Finland; anu.hangas@uef.fi (A.H.); jaakko.pohjoismaki@uef.fi (J.L.O.P.)

**Keywords:** topoisomerases, mitochondrial DNA, mitochondrial DNA maintenance, mtDNA replication

## Abstract

Like any genome, mitochondrial DNA (mtDNA) also requires the action of topoisomerases to resolve topological problems in its maintenance, but for a long time, little was known about mitochondrial topoisomerases. The last years have brought a closer insight into the function of these fascinating enzymes in mtDNA topology regulation, replication, transcription, and segregation. Here, we summarize the current knowledge about mitochondrial topoisomerases, paying special attention to mammalian mitochondrial genome maintenance. We also discuss the open gaps in the existing knowledge of mtDNA topology control and the potential involvement of mitochondrial topoisomerases in human pathologies. While Top1mt, the only exclusively mitochondrial topoisomerase in mammals, has been studied intensively for nearly a decade, only recent studies have shed some light onto the mitochondrial function of Top2β and Top3α, enzymes that are shared between nucleus and mitochondria. Top3α mediates the segregation of freshly replicated mtDNA molecules, and its dysfunction leads to mtDNA aggregation and copy number depletion in patients. Top2β, in contrast, regulates mitochondrial DNA replication and transcription through the alteration of mtDNA topology, a fact that should be acknowledged due to the frequent use of Topoisomerase 2 inhibitors in medical therapy.

## 1. Topoisomerase Types and Their Known Functions

All organisms require DNA topoisomerases to resolve sterical problems during the maintenance and expression of their genomes. Topoisomerases control DNA topology, supercoiling, and interconnection, and their action is required for the handling of long DNA molecules during replication, transcription, segregation, and recombination. All topoisomerases catalyze controlled strand breaks by transesterification between a tyrosyl group at the active center of the protein and a phosphor group of the DNA strand. After a change in the topology of the DNA molecule, this transient covalent bond between protein and DNA is broken, and the cut DNA strand is resealed [1].

Homology analysis of the known topoisomerases suggests that they have developed from five distinct ancestral enzymes [2]. Interestingly, the evolutionary relationship of topoisomerases does not follow the evolutionary lineages, but instead suggests the loss and re-acquisition of topoisomerases via lateral gene transfer among all phyla.

Topoisomerases are commonly grouped by their mechanistic properties; Type I topoisomerases create a strand break only on one of the two DNA strands and then transfer the other strand through before religation. Most topoisomerases of this type are monomeric, and they are driven by the energy of the helical torsion of the DNA template, therefore not requiring ATP for catalysis. Two sub-groups of Type I topoisomerases are known—Type IA enzymes open the strand by binding to the 5′-end of the cut site and allow a second strand to pass through. Thus, they can decatenate single-stranded DNA but also relax negative supercoils. This family is the most ancient topoisomerase group and comprises bacterial TopI and TopIII, eukaryotic Top3α, archaeal Top, as well as bacterial and archaeal reverse gyrase. The members of this topoisomerase family are mainly associated with DNA recombination and repair [3] and are found across the tree of life. Although Type IB enzymes also catalyze single strand break-mediated relaxation, they are not structurally related to Type 1A enzymes and alter DNA topology by rotation instead of strand passage. These enzymes can relax both positive and negative supercoils and bind to the 3′-end of the cut DNA. All eukaryotic TopI versions belong to this group, in addition to several bacterial enzymes, such as the thermophile TopV and the TopI of most bacteriophages [4,5]. 

In contrast to the single-strand cutting Type I topoisomerases, Type II enzymes create a full double-strand break in DNA and catalyze the passage of a second DNA duplex. These enzymes are symmetrically built multimers and bind both ends of the cut DNA. They can unwind positive and negative supercoils and decatenate entangled DNA molecules. Most known type II enzymes belong to the subgroup IIA, among them eukaryotic Top2, bacterial TopIV, and T4 phage TopII. This subgroup also contains DNA gyrases, bacterial topoisomerases with the exceptional ability to introduce positive supercoils and thus to regulate gene expression of the bacterial genome [6]. Type IIB topoisomerases, in contrast, are found mostly in archaebacteria and some higher plants. These enzymes create a two-base pair overhang during the cut [7] and include Spo11 proteins, which are present in all meiotic eukaryotes [8].

Additional to classical topoisomerases, other proteins with topoisomerase-like functions exist. The poorly characterized microrchidia proteins possess gyrase-like domains and are able to catenate and decatenate as well as relax supercoiled DNA. The members of this family are involved in chromatin remodeling and epigenetic regulation and require additional proteins for their activity [9,10]. 

### Topoisomerase Functions

While all topoisomerases catalyze the same mechanistic function, the controlled opening and resealing of DNA strands, they participate in many different aspects of DNA metabolism.

Through the regulation of supercoiled genomic regions, topoisomerases can promote or block the access and progression of replication and transcription complexes. Relaxation of positively supercoiled DNA facilitates the opening of the DNA double strand during replication and transcription. The condensation of DNA instead suppresses transcriptional access and allows chromosomal segregation.

During replication and transcription of DNA, the strand opening builds up positive supercoils in front of and negative supercoils after the replication or transcription complex that, without relaxation, would arrest the fork or bubble progression. Depending on the organism, the positive and negative supercoils are removed by either Type I or II topoisomerases.

After DNA replication of longer genomes, the daughter molecules are intertwined, and a Type II topoisomerase is required to separate them and facilitate the distribution during cell division. In ring-shaped genomes, replication can also lead to the formation of hemicatenanes, where the strands are connected by a single-strand interlock, requiring a type I topoisomerase for resolution.

Besides their various functions in DNA replication, topoisomerases participate in homologous recombination of DNA. Topoisomerases of the Spo11 family create programmed double-strand breaks to initiate homologous recombination during meiosis [11]. Although these proteins are related to archaeal topoisomerase IV, they appear to require cofactors for DNA cleavage and are therefore not typical topoisomerases in the most narrow sense [12]. 

In addition, the bona fide topoisomerases 2 and 3 participate in homologous recombination. Top2β interacts with chromatin loop anchors in the nuclear DNA, potentially facilitating the condensation and relaxation of chromatin but at the same time promoting genomic rearrangements [13]. In both bacteria and the eukaryotic nucleus, Top3α interacts with RecQ helicases that are associated with genome stability and repair of DNA double-strand breaks [14,15]. In particular, the topoisomerase/helicase complex dissolves double Holliday junctions and thus reduces the frequency of sister chromatid exchange during homologous recombination [16].

Recently, Top2β was found to be actively recruited to DNA double-strand breaks (DSB) in nuclear DNA, and its knockout impaired their repair by homologous recombination [17], indicating that this topoisomerase has an additional function in DNA repair.

## 2. Topoisomerase Requirements in Mitochondrial DNA Maintenance

All mitochondria are thought to have developed from an α-proteobacterial ancestor related to the extant *Rickettsiae* family with compact circular genomes [18]. As many genes of this ancestor have been lost or transferred into the nucleus, the mitochondrial genome of most multicellular organisms is reduced to a small, compact genome, typically encoding only for several subunits of the respiratory chain, ribosomal and transfer RNAs required for mitochondrial translation, and occasionally other proteins involved in transcription, RNA processing, or protein import [19]. 

Mitochondrial DNA (mtDNA) in yeast exists in a variety of forms. In the baker’s yeast, *S. cerevisiae,* it exists predominantly as polydisperse linear tandem arrays, and circular forms represent a minority, while in *Candida albicans*, mtDNA forms a branched network [20,21,22,23]. The much larger mitochondrial genomes of plants show even higher variability of organization, including plasmids replicating independently of the main mitochondrial genome [24]. While the mitochondrial genomes in most metazoans are circular, protists, cnidaria, and sponges have typically differently organized mitochondrial genomes, including concatenated circles, linear concatemers, or multipartite genomes, sometimes co-existing within the same species [19,25,26,27,28]. Until now, only one bilateral animal, the woodlouse *Armadillidium vulgare*, was found to have a genome existing as both linear monomer and circular dimer [29], but it is likely that this feature is more common and has been overlooked in other species.

In mammals, mtDNA typically exists mainly as small single circles in either supercoiled or relaxed conformation, and only a small fraction of the molecules is present in catenated form or as multimeric circles (see Figure 1). Human adult cardiomyocytes pose an exception to this norm, as during childhood, their mtDNA starts to form complex networks comprising many molecules connected by three- and four-way junctions [30] that co-exist with single-molecules mtDNA. This feature is unknown from other mammals but can be induced in mouse hearts through overexpression of the mtDNA maintenance proteins TWNK or TFAM. Because of the considerable variation in the mitochondrial genome organization, a range of replication mechanisms are known, ranging from theta-type replication over recombination-initiated replication to rolling circle mechanisms [31,32,33]. 

In circular genomes, the action of topoisomerases is required to relax excessive supercoiling during fork progression [34]; additionally, it might facilitate the access of transcription and replication complexes to the promoter and origin sequences and regulate the balance between transcription and replication to avoid collisions of the replication and transcription fork. During theta-type replication of a circular genome, the daughter molecules can end up intertwined as hemi- or full catenanes. The segregation of these structures requires the action of either Type I or II topoisomerases, depending on the catenane type. 

Replication initiation of linear and multidispersed mitochondrial genomes appears to depend on recombination events at homologous repeats, although the details are unknown in most cases [19,21,33,35]. Topoisomerases might be involved in the regulation of this recombination-dependent replication as they are in T4 phages [36], but for mitochondria, this has not been investigated.

An especially intricate control of topology is required for the maintenance of kinetoplast DNA, the mitochondrial genome of trypanosomid parasites that is organized as a complex network of circles. Replication of this genome requires the decatenation and separation of individual minicircles, which are then replicated by a rolling-circle mechanism and later reconnected to the network [37].

## 3. Mitochondrial Topoisomerases throughout the Eukaryotic Kingdoms

Most eukaryotes contain at least three topoisomerases for nuclear DNA maintenance, one of each belonging to the types IA, IB, and IIA. The eukaryotic TopIA is related to bacterial TopIII and is thus named Topoisomerase 3 or TOP3. This protein usually cooperates with helicases in the resolution of Holliday junctions. In mammals, the Top3 gene has been duplicated, with Top3α regulating homologous recombination during repair [40,41], while its paralogue, Top3β, instead is mainly involved in meiotic recombination [42] and has also been reported to have RNA topoisomerase activity [43]. The eukaryotic Topo IB, named Top1, can relax positive supercoils, and in the nucleus, is the main enzyme involved in the relaxation of positive supercoils accumulating ahead of replication forks and transcription bubbles. All eukaryotes also possess a Type II topoisomerase, usually Top2, which participates in replication and segregation of daughter genomes but can have additional functions. In mammals, this gene is duplicated, and while Top2α is mostly expressed in proliferating cells, Top2β is most prominent in finally differentiated cells [44,45].

Mitochondrial DNA also requires the control of topology and resolution of DNA tangles. Thus, every eukaryotic organism possesses at least one topoisomerase that is encoded by the nuclear genome and post-translationally imported into mitochondria. These topoisomerases often show dual localization, allowing the use of the enzyme for the maintenance of both mitochondrial and nuclear or mitochondrial and plastid genomes. This dual localization has made it difficult to reliably identify mitochondrial topoisomerases and to study their functions separately from the nuclear counterpart. For example, yeast such as *Saccharomyces cerevisiae* and *Schizosaccharomyces pombe* have been predicted to possess mitochondrial Top1 and Top3 [46], but Type IIA topoisomerases are still elusive. 

In photosynthetic organisms, Gyrase, Top1, and Top2 have been found in mitochondria, but not all groups possess all three [47]. Most algae, with the exception of Chlorophyta, possess a mitochondrial Top2. Instead, Chlorophyta have Top1A and sometimes also Gyrase. *Arabidopsis* mitochondria share both type I and II topoisomerases with the nucleus [48,49]. Although the precise number of mitochondrial topoisomerases in vascular plants is yet unclear [50], at least one gyrase-like topoisomerase, GyrA, is essential, as the inactivation of its gene leads to embryonic lethality [48].

Protozoans usually possess three topoisomerases of the type IA, IB, and IIA, with some, such as the apicomplexan parasite Plasmodium, also having an archaeal-type TopIV [51]. The role of topoisomerases in organelle genome maintenance is perhaps best studied in trypanosomatid parasites such as *Trypanosoma* and *Leishmania*. In these organisms, the mitochondrial (kinetoplast) genome is organized in a complex network of interlocked circles, which requires both type I and II topoisomerase activity for maintenance [52,53]. The type I topoisomerase Top1A is required in the last phases of replication, while the type II Top2βmt re-attaches freshly replicated minicircles to the network and keeps the network intact [54]. 

In *Drosophila melanogaster,* Top3α is known to localize to both nucleus and mitochondria [55], but no other topoisomerase has been found in the organelle to date. Vertebrates again contain Top1, Top2, and Top3 to fulfill the requirements of mtDNA maintenance, with two of these three topoisomerases shared between nucleus and mitochondria [56].

## 4. Mitochondrial Topoisomerases in Higher Animals

Topoisomerases in higher animals such as humans and mice are perhaps the best known of all eukaryotes because of their biomedical importance. Mammals, and likely all vertebrates, have four different mitochondrial topoisomerases, with Top1mt being the only one that exists exclusively in mitochondria. The three other Topoisomerases, Top2α, β, and Top3α, are encoded by the same genes as their nuclear counterparts, and their mitochondrial functions have been addressed only recently (for an overview, see Table 1).

### 4.1. Top1mt 

While in non-vertebrate animals, the same *TOP1* gene product seems to be shared between nucleus and mitochondria, vertebrates possess a separate gene for the mitochondrial topoisomerase Top1mt. The mitochondrial paralogue lacks most of the long N-terminal extension present in the nuclear Top1and therefore has reduced DNA binding affinity [60,61]. 

Top1mt regulates mtDNA topology by relaxing negative supercoils, thus also acting as a negative regulator of mitochondrial transcription [56,57]. Top1mt binds to the non-coding region of mtDNA and might act as a topological barrier, shifting the balance from transcription towards replication of mtDNA [62,63]. Loss of Top1mt leads to impaired mitochondrial function, increased production of oxidative radicals, and DNA damage [64]. This is probably the reason for alterations of Top1mt expression in cancer development, although it appears to depend on the type of cancer whether it is downregulation or enhanced expression of Top1mt that supports cancer development and metastasis [65,66,67].

Top1mt^−/−^ fibroblasts show decreased mitochondrial ATP production and increased oxidative damage, which cannot be compensated by upregulation of mitochondrial biogenesis [64]. Although Top1mt is thus important for normal mitochondrial function, Top1mt knockout mice are viable and relatively healthy [64], suggesting that other mitochondrial topoisomerases might compensate its loss at least partially. The importance of Top1mt becomes more apparent under stress conditions. Upon chronic exposure to doxorubicin, a Top2 inhibitor with known mitochondrial toxicity, Top1mt knockout mice show increased damage of cardiac mitochondria, loss of respiratory chain function, and increased lethality compared to wildtype mice [68]. While this deleterious effect is specific for heart tissue, and no difference was found in skeletal muscle from the same mice, a second study by the same authors showed Top1mt to be required for mtDNA replication during liver regeneration [58], indicating a general requirement for Top1mt under mtDNA stress (for a more detailed overview of the functions of Top1mt, please see the review focusing on this enzyme in the same issue).

### 4.2. Top2

Maintenance of circular genomes, including mitochondrial DNA, should require type II topoisomerases to resolve catenanes arising at the end of replication. Therefore, it is curious that very little has been known about the mitochondrial function of these topoisomerases until recently. The first attempts to identify a mitochondrial type II DNA topoisomerase were done in the late 1970s, when a bacterial gyrase-like enzyme was speculated to be present in mammalian mitochondria [69,70]. Shortly after, the activity of a type II topoisomerase was shown in rat liver and human leukemia cell mitochondria [71,72,73], but precise information about this enzyme remained elusive. Nearly twenty years later, Low and coworkers found a truncated version of nuclear Top2β in bovine heart mitochondria [74]. While this truncated version lacked the C-terminal domain, later studies, including our own, found the full length versions of both Top2α and Top2β in mitochondria of cultured cells and mouse tissues [38,56]. A third recent study failed to detect their presence in mitochondria [59], potentially due to their low abundance and sensitivity to degradation.

The eukaryotic topoisomerase 2 isoforms in vertebrates, Top2α and Top2β, are encoded by separate genes and share 68% sequence similarity, with their N-terminal part being more conserved with 78% identity [75]. As far as is known, the nuclear and mitochondrial Top2 versions seem to be identical. Unlike for Top1mt and Top3α, no mitochondrial targeting sequences have been identified, and it is still unknown how and when Top2 proteins are targeted to mitochondria.

In the nucleus, Top2α and Top2β possess distinct functions; Top2α is required for chromatin condensation and segregation in proliferating cells, whereas Top2β plays a role in differentiation and maturation as it functions, for example, in DNA repair and transcriptional activation of particular genes, especially in developing neurons [76]. Nuclear Top2α and Top2β do not only have isoform-specific functions but also different expression profiles in tissues. Top2α is present mainly in proliferating cells, whereas expression of Top2β is not dependent on the cell cycle and is most abundant in post-mitotic cells [44,45]. The mitochondrial Top2α protein seems to be proliferation-dependent, being absent in differentiated tissues and cells [38,56]. As replication is also required for mtDNA maintenance in post-mitotic cells, it is therefore likely that Top2β is the main topoisomerase 2 isoform in mitochondria. In fact, Top2α does not seem to be essential for mtDNA maintenance even in proliferating cells, since Top2α knockdown in HeLa cells does not alter mtDNA topology, copy number, nor the levels of 7S DNA bound to the non-coding region [38,59]. Interestingly, specific binding sites for Top2α have been mapped to the ends of the D-loop region, and therefore Top2α has been suggested to be involved in scaffolding this region and protecting the 7S ends from degradation [56].

In contrast to Top2α, Top2β was found to directly participate in mtDNA maintenance and to regulate mtDNA topology [38]. Top2β relieves positive topological stress, since inhibition of the enzyme leads to accumulation of positive supercoils [38]. As Top1mt relaxes negative supercoils, the two topoisomerases likely work in concert to maintain the torsional homeostasis of mtDNA. Top2β possibly also has an important function in replication, as Top2β co-localizes with replicating nucleoids, and its inhibition blocks replication initiation almost entirely [38]. Whether this effect is direct or the initiation of mtDNA replication requires a certain topological state remains to be shown.

Even if mitochondrial topoisomerases appear to have specific roles in mtDNA maintenance, the universal nature of changes in DNA topology allows the functional overlap between the different enzymes. Top2β^−/−^ mouse embryonic fibroblasts do not seem to possess any mitochondrial defects [57], in contrast to sudden knockdown or poisoning of Top2β [38]. Interestingly, Top2β levels in Top1mt^−/−^ MEFs are significantly up-regulated, indicating that after an adaptation phase, Top1mt and Top2β can substitute each other. 

### 4.3. Top3α

Topoisomerase 3 is found in mitochondria of all higher animals. The mitochondrial protein is derived from the same gene as the nuclear enzyme by using an alternative start codon producing an N-terminal mitochondrial targeting sequence [46]. As this targeting sequence is cleaved upon protein import, the active mitochondrial protein is nearly identical to its nuclear counterpart.

The knockout of mitochondrial Top3α in *Drosophila* leads to reduced mtDNA copy number, accumulation of mtDNA deletions, and declining mitochondrial function with age, decreasing the life span of the animals [77]. The germ lines of the flies seem to be specifically affected—females lacking mitochondrial Top3α lay sterile eggs with heavily depleted mtDNA levels, while males exhibit strongly reduced fertility due to impaired spermatid formation [55]. The importance of Top3α for mtDNA maintenance during gamete synthesis is also evident from its increased localization to mitochondria during later stages of the spermatid formation.

The most convincing evidence for a central role of Top3α in mtDNA maintenance comes from the fact that certain compound heterozygotic mutations in Top3α can give rise to progressive external ophthalmoplegia (PEO), a mitochondrial DNA maintenance syndrome caused by pathological mutations in the mitochondrial replicative helicase TWNK and DNA polymerase γ [59]. In addition to mtDNA deletions typical for PEO, abundant high molecular weight forms were observed in skeletal muscle of the affected patients.

As Top3α is a type I topoisomerase, it is only able to decatenate hemicatenanes arising through replication termination. The decatenation of true catenanes requires the activity of a type II topoisomerase, or alternatively, a single-strand break close to the cut site of the Type I topoisomerase. Interestingly, the catenated mtDNA molecules described in this study were sensitive to S1 nuclease, suggesting them to be partially single-stranded. In this case, the activity of a type I topoisomerase would be sufficient to resolve any interlocked circles at the single-stranded sites.

The patient mutations were shown to possess reduced topoisomerase activity in vitro, and notably, the depletion of Top3α by siRNA did not only impair the segregation of freshly replicated mtDNA molecules, but it lead to replication stalling in all regions of the genome. It is therefore possible that Top3α functions not only as a decatenase but also relieves torsional stress during replication and thus facilitates the progression of the replication fork. In this case, the observed depletion would not be caused by impaired segregation alone but by a direct inhibition of replication.

The important role of Top3α for mtDNA maintenance was confirmed by an independent study that described additional patients with mutations in *TOP3A* suffering from a Bloom’s syndrome-like disorder, which involved mtDNA depletion in skeletal muscle [16]. A summary of the different topological regulations in mtDNA metabolism and, as far as is known, the executing proteins, is depicted in Figure 2.

## 5. Inhibition of Topoisomerase Function in Mitochondria

Although topoisomerases open and reseal DNA in a strictly controlled manner, the creation of strand breaks also poses a risk for genome integrity. Chemicals inhibiting the action of topoisomerases, so-called topoisomerase poisons, can cause persistent DNA-enzyme adducts, promote genomic rearrangements, and result in cell death.

The cytotoxic properties of topoisomerase poisons can be exploited to treat bacterial and parasite infections and selectively kill cancer cells, which, due to their active nuclear DNA replication, depend heavily on topoisomerase activity. Unfortunately, the low specificity of the inhibitors and the many roles of topoisomerases unrelated to DNA replication, such as transcription and repair, impose the risk of damage to post-mitotic cells, causing severe side effects of these drugs.

Anthracyclines, a group of aromatic compounds found in Streptomyces-bacteria, are topoisomerase II poisons preventing the re-ligation of DNA after the torsional conversion, but they also intercalate directly into DNA, forming covalent DNA adducts and generating reactive oxygen species [78,79,80,81]. 

Anthracyclines possess excellent antitumor efficacy and are therefore frequently used in the majority of cancer therapies, but they also cause dose-dependent mitochondrial dysfunction in cardiomyocytes and progressive cardiomyopathy, leading to death in about 5% of treated patients [82]. This adverse effect is at least partially caused by the inhibition of Top2β, as the knockout of this enzyme is protective [83]. Doxorubicin inhibits Top2β and causes the strong accumulation of positively supercoiled DNA [38,84].

Interestingly, the poisoning of mitochondrial Top2s with doxorubicin does not only lead to topological changes and the accumulation of protein-DNA adducts, but it also leads to an increase in dimeric and trimeric mtDNA molecules [38]. In contrast to catenanes of interconnected monomeric rings, these multimeric mtDNA molecules are unicircular, single circles of two to three genomes in size, and likely arise through homologous recombination, suggesting this to be a repair pathway for the double-strand breaks caused by Top2 poisoning.

Besides anthracylines, a second group of Topoisomerase 2 poisons are known to impair mitochondria. Fluoroquinolones, originally thought to exclusively inhibit bacterial TopIV and gyrase and therefore used in antibacterial therapy, are known to also affect eukaryotic Top2s [85,86,87]. While fluoroquinolone antibiotics are prescribed and consumed frequently without problems, in rare cases, they can lead to adverse effects. The best known side effect is inflammatory tendinopathy, but in the last years, a wide range of neurological side effects ranging from neuronal pain to mental disorders have gained recognition [88]. While some risk factors predestined for these side effects are known, such as decreased renal function or co-treatment with corticosteroids, for most affected persons, it remains unclear why the antibiotic had such severe side effects [89,90].

Fluoroquinolones are known to impair mitochondrial function and cause oxidative stress [91,92,93], and in cultured cells, they can deplete mtDNA [94,95]. We found that high doses of the fluoroquinolone ciprofloxacin do inhibit mitochondrial Top2β, leading to the accumulation of supercoiled mtDNA and impaired mtDNA replication [38]. While this inhibition of replication should be released after the end of the usually short therapy, a temporary block of mtDNA replication and concomitant depletion might lead to mtDNA re-arrangements or clonal expansion of pre-existing mtDNA deletions. The concentrations necessary to cause a dramatic impairment of mtDNA replication in cell culture are rarely reached during medical therapy, but lower doses might be sufficient to disturb the balance of topoisomerase activity. Especially interesting is whether the two high frequency Top1mt variants reported to have reduced relaxing activity [66] are overrepresented in patients suffering from fluoroquinolone toxicity. 

While the impairment of mitochondrial function by fluoroquinolones is undesired during antibacterial therapy, it appears to contribute to the recently acknowledged anti-cancer activities of these drugs [86,96,97]. In lung cancer cells, the treatment with levofloxacin impaired respiratory chain activity, causing oxidative stress and apoptosis [98]. 

As with Top2 poisons, inhibitors of Top1 topoisomerases are clinically used to treat cancers. Camptothecin inhibits both Top1 and Top1mt in vitro, but the inefficient access to the mitochondrial compartment prevents the inhibition of Top1mt in vivo [99]. Lamellarin D, another specific inhibitor of Top1, instead does accumulate in mitochondria and induces Top1mt-mtDNA adducts and mtDNA damage [99].

Topoisomerase poisons are not the only cause of topoisomerase adducts. Oxidized nucleotides such as 8-oxo-guanosine can trap topoisomerase I on the DNA, which might be a frequent event in the highly oxidative mitochondrial environment [100,101]. In the nucleus, the repair of these protein-linked DNA-strand breaks requires processing by the tyrosyl-DNA-phosphodiesterases, Tdp1 and Tdp2. Tdp1 excises Top1-DNA adducts and is involved in oxidative stress repair, while Tdp2 resolves adducts of Top2 and potentially Top3 to DNA [102]. Both enzymes are also found in mitochondria, where Tdp1 removes Top1mt-DNA adducts and thus influences the balance of mtDNA transcription and replication [103,104,105,106,107].

## 6. Open Questions

While the last years have advanced our knowledge about the functions of mitochondrial topoisomerases, at least in higher animals, many questions remain.

In particular, the functional overlap and division of labor between the different mitochondrial topoisomerases is still unclear. While the knockdown of individual topoisomerases leads to aberrations and disturbances of mtDNA replication and copy number, cell and mouse models lacking topoisomerase Top1mt, Top2β, or even mtTop3α exist, suggesting that after an adaptation phase, the remaining topoisomerases can partly compensate the lack of function under normal conditions.

All topoisomerases appear to interact with the non-coding region of mtDNA [56,59,62], but for the sequence or structure specificity of this binding as well as in the case of Top2 and Top3α, the precise binding sites are unclear. 

It remains equally unsolved which topoisomerases are acting during transcription and replication to relax the differential supercoiling accumulating before and after the fork. This function can be fulfilled by both Type I and Type II topoisomerases, thus it remains to be seen which of the known enzymes in mitochondria is acting outside of the D-loop to ensure the progression of replication and transcription complexes. Top1mt has been reported to directly interact with POLRMT [57], suggesting this protein is involved in the regulation of supercoiling caused by transcription. However, overexpression of Top1mt reduces rather than increases transcription, thus a second topoisomerase might be involved. The loss of Top1mt also does not abolish mitochondrial replication completely [58], thus here a second topoisomerase must be able to relieve the torsional stress.

### 6.1. Regulatory and Accessory Factors of Mitochondrial Topoisomerases

While the mechanistic actions of topoisomerases have been studied in detail, our knowledge concerning their regulation is rather recent. In the nucleus p53, SMARCA4 and BAF were all found to recruit and stimulate Top1 and Top2 [108,109,110]. In mitochondria, such a factor could act as a sensor of mitochondrial function, conveying the requirements for transcription or replication of mtDNA into topology changes. p53 is suggested to be involved in mtDNA genome maintenance [111]. It also serves as a sensor of oxidative balance and oxygen and regulates mitochondrial metabolism, thus is an ideal candidate for the adjustment of mtDNA maintenance to mitochondrial activity [112].

Additionally, DNA damage resulting from excessive oxidative stress might regulate the activity of topoisomerases, as Top2 was shown to be especially recruited to oxidized or alkylated nucleotides and abasic sites in nuclear DNA [113]. In mitochondria, this specific recruitment to damaged DNA molecules could initiate mtDNA repair processes or, in contrast, suppress transcription and replication of faulty genomes. 

### 6.2. Recombination of mtDNA

While the participation of topoisomerases in homologous recombination in the nucleus is well established, not much is known about mitochondrial recombination and the factors involved in its mechanism. Homologous recombination was long thought to exist only in mitochondria of yeasts and plants [114,115], but over the years, more and more evidence of this repair pathway has been found in mitochondria of higher animals [116]. In humans, four-way-junctional mtDNA molecules likely representing recombination intermediates increase dramatically during the postnatal hypertrophic growth of the heart, suggesting this to be an adaptation to the increased oxidative stress in this phase and homologous recombination in a repair pathway potentially dealing with mtDNA strand breaks [117,118].

In the nucleus, Top3α cooperates with Rmi1 and RecQ helicases to regulate homologous recombination [41]. Interestingly, a RecQ helicase, RecQ4, is also found in mitochondria, and other nuclear helicases involved in nuclear DNA repair are present, namely SUV3, DNA2, and PIF1 [119]. Even the replicative mitochondrial helicase TWNK was found to have strand-exchange activity [120] and therefore might be a core component of a mitochondrial recombination machinery [121].

Whether Top3α interacts with any of these helicases to regulate mitochondrial recombination is currently unclear, but at least both Top3α and TWNK are present in high levels in tissues exhibiting prominent mtDNA recombination intermediates [38,122]. Alternatively, Top2β might act during homologous recombination repair, as it does in the nucleus [17]. The fact that multimeric mtDNA molecules increase upon Top2 inhibition by doxorubicin or ciprofloxacin suggests that Top2β could have a suppressive role for recombination [38]. Alternatively, Top2 poisoning might cause the observed increase of mtDNA dimers via the accumulation of linearized mtDNA, serving as the substrate for recombination without the direct involvement of the topoisomerase itself.

While the topics mentioned above should be addressed to deepen our basic understanding of mitochondrial DNA maintenance, there is also urgent need to widen our knowledge of the medical aspects of topoisomerase inhibition.

### 6.3. Molecular Mechanisms of Delayed Mitotoxicity

Topoisomerase inhibitors are valuable medical drugs in the fight against cancer and bacterial infections. However, the effects of topoisomerase poisoning on genome integrity, especially in mitochondria of non-proliferating cells, is poorly understood. 

Both fluoroquinolones and anthracyclines cause an acute impairment of mitochondrial function, but the molecular reasons for the progressive problems, often arising only after the end of the therapy, are yet unclear. Does the inhibition of mitochondrial Top2 lead to extensive mtDNA deletion formation? Or does it, by the creation of a bottleneck situation, cause the clonal expansion of already existing deletions? Can the mitochondrial damage caused by topoisomerase poisons be somehow reverted or healed once the inhibitor is removed? Similar questions have been present in the field of mtDNA damage diseases for several decades, thus any advance would find a wide audience.

### 6.4. The Role of Mitochondrial Topoisomerases in Cancer Biology

From the medical perspective, one of the outstanding questions is the connection of mitochondrial topoisomerases with cancer development and progression.

The mitochondrial activity of topoisomerases can influence the development and progression of cancers at many stages.

On one hand, the loss of Top1mt is known to cause mitochondrial dysfunction, elevated oxidative stress, and an increase in both mitochondrial and nuclear DNA damage [64], thus increasing the risk of pathogenic mutations contributing to carcinogenesis. On the other hand, elevated mtDNA copy number increases the risk of cancer formation [123], potentially by satisfying the intensive metabolic demands of proliferating cancer cells, and many cancers do exhibit overexpression of Top1mt [67]. Downregulation of Top1mt again was found to promote migration and thus metastasis of gastric cancer cells [65]. Thus, it appears that, depending on the stage, both decreased and increased activity of Top1mt might promote cancer progression and survival. Interestingly, several variants of Top1mt with altered activity are overrepresented in cancer patients [66], but it remains to be elucidated at which stage of cancer development these variants might be influential. 

Currently, no data exist about the mitochondrial protein levels of Top2α, Top2β, and Top3β in cancer cells compared to other cell types, but as these topoisomerases participate in mtDNA replication and expression, they should influence the metabolism of cancer cells.

## 7. Concluding Remarks

After a long time of ignorant bliss, mitochondrial topoisomerases have slowly gained more attention, and recent studies have advanced our understanding of their roles in mammalian mitochondrial DNA metabolism. Still, the picture is patchy, as little is known about the division of labor between the four mitochondrial topoisomerases known from mammals, their regulation under physiological conditions, or mitochondrial topoisomerase functions in other organisms. In addition, the consequences of topoisomerase dysfunction in mitochondria, whether caused by chemical inhibition or genetic mutations, should be investigated in detail to enlighten their role in human pathologies. 

## Figures and Tables

**Figure 1 ijms-20-02041-f001:**
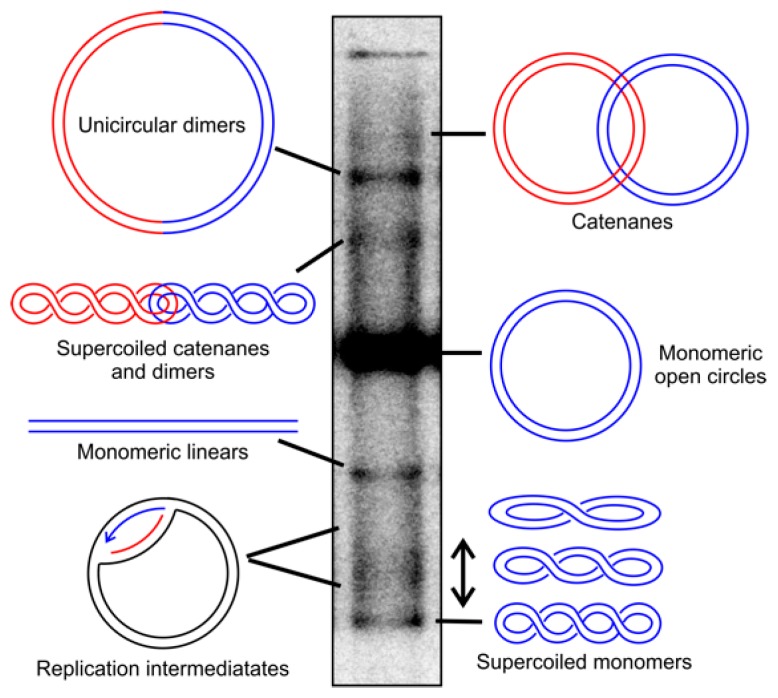
Topological conformations of human mitochondrial DNA (mtDNA). The different topological conformations of mtDNA can be separated by agarose gel electrophoresis in the absence of DNA-intercalating dyes and visualized by Southern blotting. Supercoiled mtDNA molecules can be converted into relaxed forms by treatment with Top1, while catenanes can be separated by TopIV. The treatment with restriction enzymes in a single recognition site in mtDNA creates monomeric linears of all circular molecules, as well as dimers. Replication intermediates decrease in cells treated with ciprofloxacin due to the reduced replication initiation, while replication stalling leads to their accumulation. Blue and red colors are used to visualize individual molecules, and in the case of dimers, the typical genome length of mtDNA. For the detailed treatment conditions and visualization, please see [38,39].

**Figure 2 ijms-20-02041-f002:**
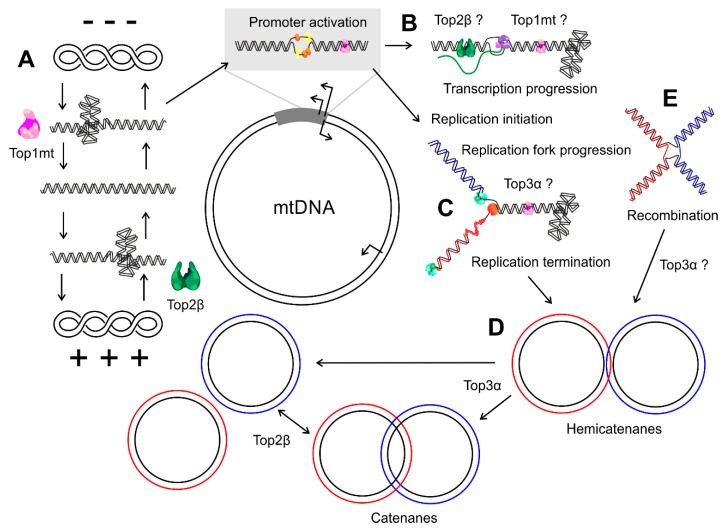
Topological changes in mtDNA metabolism. (**A**) Top1mt and Top2β together control the topological balance of mitochondrial DNA molecules, thus regulating the initiation of transcription and replication. (**B**) During mtDNA transcription, positive and negative supercoils arise in front of and behind the progressing polymerase complex, respectively, and both Top1mt and Top2β might be involved in its resolution. (**C**) Which enzymes relieve the positive supercoils accumulating in front of the replication fork and the intertwining of the daughter strands is currently unknown, but first indications speak for a role of Top3α. (**D**) At the end of replication, the replicated molecules can form hemi- or full catenanes that depend on Top3α for their resolution and segregation. (**E**) Whether Top3α or any other topoisomerase participates in the regulation of mtDNA recombination is still unclear. Blue and red colors indicate the nascent leading and lagging strand and, in the case of recombination, the two parental molecules.

**Table 1 ijms-20-02041-t001:** Features of the four topoisomerases in mammalian mitochondria.

Features	Top1mt	Top2α	Top2β	Top3α
Classification	Type IB	Type IIA	Type IIA	Type IA
Shown mitochondrial function	Regulation of transcription [56,57] and replication [58]		Regulation of replication [38]	Decatenation of hemicatenates [59]
Potential mitochondrial functions	Regulation of translation [59]	Scaffolding of the non-coding region, 7S protection [56]	Decatenation, recombination	Recombination, replication
Encoding details	Mitochondria-specific gene [60]	Probably identical to nuclear protein	Probably identical to nuclear protein	Alternative start codon [46]
Mitochondrial targeting sequence	Yes [60]	Unknown	Unknown	Yes [46]
Enzyme structure	Monomer	Homodimer	Homodimer	Monomer
Cofactors	Stimulated by Mg^2+^ or Ca^2+^ [60]	Mg^2+^, ATP	Mg^2+^, ATP	Mg^2+^
Protein size	70 kDa	174 kDa	180 kDa	110 kDa

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
