# Peer review of "Twist and Turn—Topoisomerase Functions in Mitochondrial DNA Maintenance"

_ijms, 2019, doi:10.3390/ijms20082041_

Round 1

Reviewer 1 Report

In this paper the authors review topoisomerase functions in mitochondria, with a focus on mammalian mitochondria. They start with an general introduction on topoisomerase classes and their function in the nucleus. In a second time, they introduce mitochondrial topoisomerases through the eukaryotic kingdom and then focus on mitochondrial topoisomerases in higher animals. They describe the four topoisomerases that have so far been found in mammalian mitochondria (Top1mt, Top2a, Top2β, and Top3α) and present their known or putative roles in mtDNA metabolism. Finally, they discuss the different inhibitors of topoisomerases, mainly focusing on the poisoning of Top2b.

Overall I think that the paper is well organized and rather complete. I have a few comments and suggestions:

line 100-112: maybe it would be interesting to mention the structure of yeast mtDNA, as topoisomerases present in yeast are mentioned later in the paper (line 174).

line 116 : This sentence is to my opinion too affirmative. The authors refers here to their own paper (ref #26) and there are so far no other evidences suggesting that this is an alternative replication mode. The authors should write that this is suggested and that the exact reason behind this observations are currently not known.

Line 192: the authors should mention that the mitochondrial localization of Top2a and Top2β is debated.

Line 415: This is also debated, so the authors should write “suggested” instead of “already known” Line 424: MtDNA recombination in human cells is highly debated and it seems to be more and more evidences pointing against it. The authors should discussed this controversy in this paragraph.

Paragraph 5 inhibition of topoisomerase function in mitochondria : this paragraph is rather long and focus almost mainly on inhibitors of Top2b. Knowing that the presence of this protein in mitochondria is debated, I would suggest to shorten the paragraph.

Figure 1: It would be interesting to have a more detailed explanation of the analysis of the different topological forms.

Figure 2: The legend of this figure could be clearer, with explanations for each part of the figure.

Table 1: according to the title of ref #63 Top1mt has also a role in translation?

Author Response

Many thanks for your comments and suggestions. We accommodated most of them, but do disagree with the comments about the doubtfulness of mitochondrial recombination and the mitochondrial localization of Top2b. Please find below our arguments for this choice.

line 100-112: maybe it would be interesting to mention the structure of yeast mtDNA, as topoisomerases present in yeast are mentioned later in the paper (line 174).

We separated the description of yeast, plant and lower metazoan mtDNA structure now.

line 116 : This sentence is to my opinion too affirmative. The authors refers here to their own paper (ref #26) and there are so far no other evidences suggesting that this is an alternative replication mode. The authors should write that this is suggested and that the exact reason behind this observations are currently not known.

We do agree that the reason for the mtDNA structure in human adult hearts is not solved, but here we do state here that mtDNA in this organ is organized in complex networks, without describing any easons for their arisal. These high-molecular weight complexes have been shown by electron microscopy in the cited paper and although we have been facing quite some discussions about how these mtDNA networks arise, nobody until now has actually questioned their existence. We anyways realized we should stress that these networks do contain only a proportion of the mtDNA pool and there are also single circles, so we rephrased the sentence (hopefully) appropriately.

Line 192: the authors should mention that the mitochondrial localization of Top2a and Top2β is debated.

Mitochondrial Top2a and b have been found by three independent groups especially focusing on these enzymes (Low et al. 2003 (Top2b), Zhang et al, 2014 (Top2a and b) and Hangas et al (Top2a and b), and in at least one mtDNA nucleoid proteomics study (Han et al. (2017), Proximity Biotinylation as a Method for Mapping Proteins Associated with mtDNA in Living Cells. Cell Chem Biol. 24(3): 404–414 - Top2a), so there is quite some evidence for their existence in mitochondria, although the study of Nicholls et al. (2018) fails to find them.

We do mention this fact and a potential reason for it in the later paragraph describing the current knowledge about mitochondrial Top2. In this paragraph here we would like to avoid a lengthy discussion about this fact, but we also find it inappropriate to state that the presence of Top2 in mitochondria is unclear, if it is doubted by only one study.

Line 415: This is also debated, so the authors should write “suggested” instead of “already known”

We have changed the sentence to be more cautious.

Line 424: MtDNA recombination in human cells is highly debated and it seems to be more and more evidences pointing against it. The authors should discussed this controversy in this paragraph.

We do in contrast notice that the evidence for mitochondrial recombination is increasing, but as we know the intense discussions prevailing in the field we have on purpose phrased this paragraph quite carefully. We state that not much is known, but that there is evidence for mitochondrial recombination – and evidence there is:

1.     Zsurka G, Kraytsberg Y, Kudina T, Kornblum C, Elger CE, Khrapko K, Kunz WS (2005). Recombination of mitochondrial DNA in skeletal muscle of individuals with multiple mitochondrial DNA heteroplasmy. Nat. Genet. 37:873–877.

2.     Hoolahan AH, Blok VC, Gibson T, Dowton M (2012). Evidence of animal mtDNA recombination between divergent populations of the potato cyst nematode Globodera pallida. Genetica 140:19–29.

3.     Fukui H, Moraes CT (2009). Mechanisms of formation and accumulation of mitochondrial DNA deletions in aging neurons. Hum. Mol. Genet. 18:1028–1036.

4.     Kraytsberg Y, Schwartz M, Brown TA, Ebralidse K, Kunz WS, Clayton DA, Vissing J, Khrapko K (2004). Recombination of human mitochondrial DNA. Science 304:981.

5.     Piganeau G, Gardner M, Eyre-Walker A (2004). A broad survey of recombination in animal mitochondria. Mol. Biol. Evol. 21:2319–2325.

6.     D'Aurelio M, Gajewski CD, Lin MT, Mauck WM, Shao LZ, Lenaz G, Moraes CT, Manfredi G (2004). Heterologous mitochondrial DNA recombination in human cells. Hum. Mol. Genet. 13:3171–3179.

7.     Mita S, Rizzuto R, Moraes CT, Shanske S, Arnaudo E, Fabrizi GM, Koga Y, DiMauro S, Schon EA (1990). Recombination via flanking direct repeats is a major cause of large-scale deletions of human mitochondrial DNA. Nucleic Acids Res. 18:561–567.

Even a recent paper by the group of Maria Falkenberg, previously strongly doubting mitochondrial recombination, is providing evidence of a replication-connected mechanism of deletion formation the authors call “copy-choice recombination” (Persson Ö, Muthukumar Y, Basu S, Jenninger L, Uhler JP, Berglund AK, McFarland R, Taylor RW, Gustafsson CM, Larsson E, Falkenberg M (2019). Copy-choice recombination during mitochondrial L-strand synthesis causes DNA deletions. Nat Commun. 10(1):759). Although our and many others’ idea of mitochondrial recombination is very different from the mechanism described in this in vitro study, it is supporting the existence of mitochondrial recombination in one way or another.

A detailed discussion of the pros and cons of mitochondrial recombination we do find inappropriate for this review, as the only point we would like to make here is that if mitochondrial recombination exists, topoisomerases might participate in this mechanism, as they do in so many other genomic systems.

Paragraph 5 inhibition of topoisomerase function in mitochondria : this paragraph is rather long and focus almost mainly on inhibitors of Top2b. Knowing that the presence of this protein in mitochondria is debated, I would suggest to shorten the paragraph.

As above – the presence of Top2 in mitochondria is debated by one study alone, while there is plenty of evidence that topoisomerase 2 inhibitors are very harmful for mitochondrial DNA. This long-existing knowledge has been gaining a lot of attention in the last years also outside of the science community and lead to e.g. heavy restrictions in the medical use of fluoroquinolones. For this reason we chose this as one of the central topics of this review and discuss the known inhibitors and their mitochondrial effect in detail.

To our knowledge Lamellarin D is the only mitochondria-permeable inhibitor of Top1, and this will be presented in more detail in the review by Yves Pommier in the same special edition, so we aimed to keep its discussion here rather short. We’d love to include also some data about inhibitors of Top3, but unfortunately there seems to be none existing.

Figure 1: It would be interesting to have a more detailed explanation of the analysis of the different topological forms.

We included in to the figure legends the various enzymes used to identify the topological forms. We think it’s not necessary to include the pictures of these treatments, as this has been published several times before and would require a lengthy method description.

Figure 2: The legend of this figure could be clearer, with explanations for each part of the figure.

We included now letters for each part of the figure and refer in the legend to the appropriate letter. We hope this makes the figure easier to read.

Table 1: according to the title of ref #63 Top1mt has also a role in translation?

Thanks for this notification, the mentioning of translation as a role of Top1mt got lost during some reshuffling in the table. We decided against the regulation of translation as a proven function of Top1mt, as the evidence in the cited study is not very strong yet (alterations in translation upon loss of Top1mt and coprecipitation of Top1mt with mitoribosomal subunits), but we feel the appropriate status would be “potential function”.

Reviewer 2 Report

This is a review on the mechanisms of mtDNA maintenance focusing on the roles of topoisomerases in this process. The manuscript is well written and comprehensive pointing the reader towards important literature in the field. There is a good summary of the importances of topoisomerases in mtDNA maintenance in mammals accompanied by a useful table that summarizes current knowledge on these enzymes. I make a few minor suggestions to improve some aspects of the manuscript. 

Minor points :

1.       At lines 56-57, it should read passage of a second DNA duplex (not necessarily another DNA molecule).

2.       In Table 1, the abbreviation NCR (non-coding regions) is not defined in the legend or in the rest of the manuscript.

3.       Line 96, it would be more specific to say reduces the frequency of crossovers or sister chromatid exchanges during homologous recombination.

4.       In the general section on topoisomerase functions, recent insights into the roles of topoisomerases in nuclear genome organization and chromosomal translocations done in the laboratory of AndrĂ© Nussenzweig could be discussed (Canela et al. Cell 2018).

Typos and grammatical corrections :

Line 50, Although Type IB enzymes also catalyze… , they are not structurally related to …

Line 81, would arrest fork or bubble progression.

Line 85, In ring-shaped genomes, replication can also… , where the strands are connected…

Line 87, Besides their various functions in DNA replication, topoisomerases also participate…

Line 88, Spo11 family create…

Line 157, …, while its paralogue Top3b

Line 167, Mitochondrial DNA also requires…

Line 230, detailed overview of the functions

Line 351, Besides anthracyclines, a second group…

Lines 355-356, but in the last years, a wide range of neurological…

Line 367, but lower doses might be…

Line 378, induces Top1mt-DNA adducts and mtDNA damage.

Line 379, topoisomerase adducts; oxidized nucleotides…

Line 391, … is yet unclear: while the knockdown…

Line 411, our knowledge concerning their regulation is rather recent: in the nucleus p53…

Line 434, also found in mitochondria, and other nuclear helicases involved in nuclear DNA repair are present…

Line 440, Alternatively, Top2B…

Line 447, While the topics mentioned above…

Line 456, … lead to extensive mtDNA deletion…

Line 468, On the other hand, elevated mtDNA copy number…

Line 480, 7. Concluding remarks

Author Response

Dear Reviewer 2,

many thanks for the helpful comments. We included all suggestions and corrections into the manuscript. The function of Top2b in nuclear genome organization and the risks connected with this are now included. Many thanks also for the careful proofreading of our manuscript!

Reviewer 3 Report

This is a lovely review of the current state of knowledge of topoisomerase function in the maintenance of the mitochondrial genome. It is comprehensive, and does a nice job of pointing to where we need further research. 

The English is a bit odd in places and could do with the attention of a native or fluent English speaker. Nothing is rendered incomprehensible by the language issues, but they distract from an otherwise excellent paper.

Having three paragraphs in an abstract is a bit odd.

Author Response

Dear Reviewer 3,

many thanks for the encouraging words and the reviewing of our manuscript. We had our Australian colleague proof-read the manuscript now, and we moved the last abstract sentence to the first paragraph. Hopefully the manuscript is now easier to read.